# Harmonizing corporate carbon footprints

Lena Klaaßen [1,2,3✉] & Christian Stoll [3,4✉]

Global greenhouse gas emissions need to reach net-zero around mid-century to limit global warming to 1.5 °C. This decarbonization challenge has, *inter alia*, increased the political and societal pressure on companies to disclose their carbon footprints. As a response, numerous companies announced roadmaps to become carbon neutral or even negative. The first step on the journey towards carbon neutrality, however, is to quantify corporate emissions accurately. Current carbon accounting and reporting practices remain unsystematic and not comparable, particularly for emissions along the value chain (so-called scope 3). Here we present a framework to harmonize scope 3 emissions by accounting for reporting inconsistency, boundary incompleteness, and activity exclusion. In a case study of the tech sector, we find that corporate reports omit half of the total emissions. The framework we present may help companies, investors, and policy makers to identify and close the gaps in corporate carbon footprints.

[1] TUM School of Management, Technical University of Munich, Munich, Germany. [2] Climate Finance and Policy Group, Department of Humanities, Social and Political Sciences, Swiss Federal Institute of Technology, ETH Zurich, Switzerland. [3] MIT Center for Energy and Environmental Policy Research, Massachusetts Institute of Technology, Cambridge, USA. [4] TUM Center for Energy Markets, TUM School of Management, Technical University of Munich, Munich, Germany. ✉email: lena.klaassen@tum.de; christian.stoll@tum.de

Global greenhouse gas (GHG) emissions need to reach net-zero around mid-century to limit global warming to 1.5 °C[1]. This decarbonization challenge has, *inter alia*, increased the political and societal pressure on companies to disclose their GHG emissions, and urged climate action as a top priority for internal and external stakeholders[2]. As a response, major companies—particularly from the tech sector—recently announced to become carbon neutral, or even carbon negative[3–7].

The first step on the journey towards corporate carbon neutrality is to quantify the current level of emissions accurately. In absence of binding regulation, alliances of non-governmental organizations have shaped corporate carbon accounting practices. The World Resources Institute and the World Business Council on Sustainable Development set the global standard for corporations to assess their carbon footprint with the so-called 'GHG Protocol'[8]. The GHG Protocol distinguishes three categories of emissions: scope 1 refers to direct emissions from a company's own activities, scope 2 refers to emissions from the production of purchased energy, and scope 3 refers to emissions from up- and downstream activities along the value chain[9].

For most industries in the United States (U.S.) and China, scope 3 emissions account for over 80% of the total emissions[10,11], and the share has grown globally over the past decades[12]. Although previous studies identify sources of error in scope 3 estimates[13–17], quantitative analyses remain scarce and little is known about the type and size of error. One study focusing on large U.S. companies, for instance, finds that companies on average reported less than 25% of their upstream scope 3 emissions in 2013[18].

Here we show that emission data disclosed in corporate reports omit half of the total emissions. Applying the framework we present in this study to quantify scope 3 emissions in a standardized way to a sample of 56 tech companies, we find a total gap between reported and harmonized emissions of 391 megatons (Mt) carbon dioxide equivalents ($CO_2e$) per annum. 202 $MtCO_2e$ thereof result from omitted upstream emissions and 189 $MtCO_2e$ from omitted downstream emissions. On the industry level, we find similar deviations between harmonized and self-reported carbon footprints: for IT software and service companies in our sample +99%, and for technology hardware and equipment companies +110%. On the firm level, emissions increase in the median by a factor of four through the harmonization, with deviations ranging from +0.06% to a factor of +185× in one case. The current lack of methodological clarity impedes effective carbon management strategies, hinders reduction target setting, and decreases the informative value for stakeholders.

## Results

**Accounting and reporting of corporate emissions.** The GHG Protocol reflects the most widely used framework for corporate carbon accounting[8]. The framework distinguishes three types of emissions: Scope 1 refers to direct emissions from owned or controlled sources, scope 2 refers to emissions from the generation of purchased electricity, and scope 3 refers to all other indirect emissions from up- and downstream activities along the value chain. To enable consistent and transparent reporting of scope 3 emissions, the GHG Protocol specifies 15 distinct categories up- and downstream in the value chain of the reporting company as listed in Table 1[19]. For each category, the GHG Protocol provides a minimum boundary in order to standardize which activities should be included.

Voluntary corporate carbon reporting standards and frameworks complement the GHG Protocol with the aim to ensure consistency, reliability, and completeness. Prominent examples are the Global Reporting Initiative (GRI) standards, the Sustainability Accounting

Standards Board (SASB) standards, and the International Integrated Reporting (IR) framework provided by the International Integrated Reporting Council (IIRC). While such standards and frameworks set the foundation for more comprehensive and consistent sustainability reporting, their approaches towards scope 3 disclosure remain inconclusive.

The GRI, for instance, provides standards for the reporting of economic, environmental, and social impacts, which include a dedicated standard for GHG emissions. This GRI standard 305 recognizes the importance of including scope 3 emissions and recommends the GHG Protocol's scope 3 standard for accounting and disclosure[20]. Still, companies are not required to disclose their full or most material scope 3 emissions to be GRI-compliant. The same applies to the SASB standards, which contain industry-specific guidelines to account for sustainability topics. Regarding GHG emissions, the SASB standards only comprise scope 1 disclosure for 22 out of 77 industries, without requiring scope 2 and 3 disclosures at all[21]. Likewise, the IR framework aims to guide corporate disclosure by combining financial and non-financial areas in order to highlight coherences and interdependencies. The framework, however, does not specify which types of GHG emission to report and remains silent on scope 3 emissions[22].

Besides corporate reports, thousands of companies have disclosed their environmental impact through the CDP (previously Carbon Disclosure Project). The CDP collects information from questionnaires that companies can submit on a voluntary basis[23]. The resulting reports of the CDP follow the structure provided by the GHG Protocol framework to report corporate carbon footprints. Although data needs to be handled carefully, as it is purely self-reported by companies, CDP is a comprehensive database for climate-related corporate actions and represents a key source for corporate sustainability indices.

As investors try to understand and manage their climate risks, financial data providers have created indices to benchmark corporate carbon exposure. MSCI, for instance, builds on CDP data and data from company reports in order to evaluate the weighted average carbon intensity of over 15,000 indices globally[24]. The definition of carbon intensity, however, excludes scope 3 emissions, and MSCI only divides the sum of scope 1 and scope 2 emissions by corporate sales. Others have started to include scope 3 emissions at least partially. Trucost, the data provider of S&P Carbon Efficiency Indices, for instance, accounts for the emissions from first-tier suppliers in addition to scope 1 and scope 2 emissions[25]. Indices such as the S&P Dow Jones Sustainability Index, however, resort to ESG scores based on industry-specific questionnaires or use publicly available information to select suitable companies instead of requiring uniform carbon measurement. Still, scope 3 data is not directly incorporated in the S&P indices although disclosure is queried and acknowledged[26,27].

**Three sources for error and how to overcome them.** Previous literature identifies multiple sources of error in publicly disclosed scope 3 emissions. We cluster these in three areas, which are reporting inconsistency, boundary incompleteness, and activity exclusion.

First, companies report scope 3 emissions inconsistently across different communication channels. Depoers et al.[14] find that French companies disclose lower total GHG emission figures in their corporate reports (CRs) than to the Carbon Disclosure Project (CDP). The reason for the discrepancy can be found in partially or completely omitted scope 3 emissions, which suggest that companies intentionally understate scope 3 emissions in CRs. Since the full range of responses is only shared with CDP's

**Table 1 Overview of scope 3 categories and minimum boundaries as stated in the GHG Protocol[19].**

| Scope 3 category | Category description | Minimum boundary |
|---|---|---|
| 1 Purchased goods and services | Extraction, production, and transportation of goods and services purchased or acquired by the reporting company in the reporting year, not otherwise included in Categories 2–8 | All upstream (cradle-to-gate) emissions of purchased goods and services |
| 2 Capital goods | Extraction, production, and transportation of capital goods purchased or acquired by the reporting company in the reporting year | All upstream (cradle-to-gate) emissions of purchased capital goods |
| 3 Fuel- and energy-related activities (not included in scope 1 or scope 2) | Extraction, production, and transportation of fuels and energy purchased or acquired by the reporting company in the reporting year, not already accounted for in scope 1 or scope 2, including:<br>a. Upstream emissions of purchased fuels (extraction, production, and transportation of fuels consumed by the reporting company)<br>b. Upstream emissions of purchased electricity (extraction, production, and transportation of fuels consumed in the generation of electricity, steam, heating, and cooling consumed by the reporting company)<br>c. Transmission and distribution (T&D) losses (generation of electricity, steam, heating and cooling that is consumed (i.e., lost) in a T&D system)—reported by end user<br>d. Generation of purchased electricity that is sold to end users (generation of electricity, steam, heating, and cooling that is purchased by the reporting company and sold to end users)—reported by utility company or energy retailer only | a. For upstream emissions of purchased fuels: All upstream (cradle-to-gate) emissions of purchased fuels (from raw material extraction up to the point of, but excluding combustion)<br>b. For upstream emissions of purchased electricity: All upstream (cradle-to-gate) emissions of purchased fuels (from raw material extraction up to the point of, but excluding, combustion by a power generator)<br>c. For T&D losses: All upstream (cradle-to-gate) emissions of energy consumed in a T&D system, including emissions from combustion<br>d. For generation of purchased electricity that is sold to end users: Emissions from the generation of purchased energy |
| 4 Upstream transportation and distribution | Transportation and distribution of products purchased by the reporting company in the reporting year between a company's tier 1 suppliers and its own operations (in vehicles and facilities not owned or controlled by the reporting company)<br>Transportation and distribution services purchased by the reporting company in the reporting year, including inbound logistics, outbound logistics (e.g., of sold products), and transportation and distribution between a company's own facilities (in vehicles and facilities not owned or controlled by the reporting company) | The scope 1 and scope 2 emissions of transportation and distribution providers that occur during use of vehicles and facilities (e.g., from energy use)<br>Optional: The life cycle emissions associated with manufacturing vehicles, facilities, or infrastructure |
| 5 Waste generated in operations | Disposal and treatment of waste generated in the reporting company's operations in the reporting year (in facilities not owned or controlled by the reporting company) | The scope 1 and scope 2 emissions of waste management suppliers that occur during disposal or treatment<br>Optional: Emissions from transportation of waste |
| 6 Business travel | Transportation of employees for business-related activities during the reporting year (in vehicles not owned or operated by the reporting company) | The scope 1 and scope 2 emissions of transportation carriers that occur during use of vehicles (e.g., from energy use)<br>Optional: The life cycle emissions associated with manufacturing vehicles or infrastructure |
| 7 Employee commuting | Transportation of employees between their homes and their worksites during the reporting year (in vehicles not owned or operated by the reporting company) | The scope 1 and scope 2 emissions of employees and transportation providers that occur during use of vehicles (e.g., from energy use)<br>Optional: Emissions from employee teleworking |
| 8 Upstream leased assets | Operation of assets leased by the reporting company (lessee) in the reporting year and not included in scope 1 and scope 2—reported by lessee | The scope 1 and scope 2 emissions of lessors that occur during the reporting company's operation of leased assets (e.g., from energy use)<br>Optional: The life cycle emissions associated with manufacturing or constructing leased assets |
| 9 Downstream transportation and distribution | Transportation and distribution of products sold by the reporting company in the reporting year between the reporting company's operations and the end consumer (if not paid for by the reporting company), including retail and storage (in vehicles and facilities not owned or controlled by the reporting company) | The scope 1 and scope 2 emissions of transportation providers, distributors, and retailers that occur during use of vehicles and facilities (e.g., from energy use)<br>Optional: The life cycle emissions associated with manufacturing vehicles, facilities, or infrastructure |
| 10 Processing of sold products | Processing of intermediate products sold in the reporting year by downstream companies (e.g., manufacturers) | The scope 1 and scope 2 emissions of downstream companies that occur during processing (e.g., from energy use) |

**Table 1 (continued)**

| Scope 3 category | Category description | Minimum boundary |
|---|---|---|
| 11 Use of sold products | End use of goods and services sold by the reporting company in the reporting year | The direct use-phase emissions of sold products over their expected lifetime (i.e., the scope 1 and scope 2 emissions of end users that occur from the use of: products that directly consume energy (fuels or electricity) during use; fuels and feedstocks; and GHGs and products that contain or form GHGs that are emitted during use)<br>Optional: The indirect use-phase emissions of sold products over their expected lifetime (i.e., emissions from the use of products that indirectly consume energy (fuels or electricity) during use) |
| 12 End-of-life treatment of sold products | Waste disposal and treatment of products sold by the reporting company (in the reporting year) at the end of their life | The scope 1 and scope 2 emissions of waste management companies that occur during disposal or treatment of sold products |
| 13 Downstream leased assets | Operation of assets owned by the reporting company (lessor) and leased to other entities in the reporting year, not included in scope 1 and scope 2—reported by lessor | The scope 1 and scope 2 emissions of lessees that occur during operation of leased assets (e.g., from energy use)<br>Optional: The life cycle emissions associated with manufacturing or constructing leased assets |
| 14 Franchises | Operation of franchises in the reporting year, not included in scope 1 and scope 2—reported by franchisor | The scope 1 and scope 2 emissions of franchisees that occur during operation of franchises (e.g., from energy use)<br>Optional: The life cycle emissions associated with manufacturing or constructing franchises |
| 15 Investments | Operation of investments (including equity and debt investments and project finance) in the reporting year, not included in scope 1 or scope 2 | A reporting company's scope 3 emissions from investments are the scope 1 and scope 2 emissions of investees (proportional share of investment in the investee) |

investor signatories, companies may withhold more comprehensive emission data from the general public[14]. This behavior might be reinforced by the evaluation scheme of the CDP, which openly communicates scores without indicating emission figures. In the evaluation process, the CDP disregards information outside the program responses and there is no obligation to provide consistent information in CRs[28]. Hence, a good score may improve a company's publicly perceived credibility with regard to the quality and completeness of their disclosures—despite reporting inconsistently across channels. This can also apply to high emitting companies as the CDP scoring system aims to provide an indication of a company's level of action to assess and manage its environmental impact instead of its level of sustainability[28].

Second, emission calculations of scope 3 categories partly face incompleteness with regard to the minimum boundaries set by the GHG Protocol. The GHG Protocol's scope 3 standard recommends companies to choose the most suitable calculation approach for each of the 15 scope 3 categories depending on data availability and quality[29]. The proposed methods can be traced back to three basic carbon accounting approaches: economic input-output, process-based, or a hybrid of the two. Economic input-output analysis is a top-down technique that uses financial transaction data. Combined with emission factors, this method enables straightforward and system-complete emission calculations[30]. In contrast, process-based analysis is a bottom-up technique that uses detailed estimations of each step[31]. A hybrid model starts with a bottom-up estimate and fills the gaps with top-down figures[32]. To enhance specificity, companies are encouraged to draw on primary data for categories which are highly influential[19]. The CDP fosters primary data collection for upstream emissions through its 'Supply Chain Program', which contains emissions data of over 5,500 tier 1 suppliers of 115

member companies. However, only one third of the suppliers reports own scope 3 emissions[33]. As a consequence, most companies cannot quantify the emissions along their entire supply chain with primary data only, which results in boundary incompleteness if the gaps are not filled with secondary data.

Third, reporting companies may neglect relevant scope 3 activities entirely. Although the GHG Protocol's scope 3 supplement provides guidance for companies, the supplement falls far short of meeting the acceptance of the basic standard[13]. The CDP structures its questionnaire along the 15 scope 3 categories but leaves it to the participants to identify relevant categories (see supplementary data: sheet 4.3). It is estimated that two categories alone, purchased goods and services (category 1) and use of sold products (category 11), together account for almost the entire scope 3 emissions[34]. Still, across industries, the relative importance of categories appears to differ. The share that the categories 1 and 11 capture varies between 25% (electric utilities & independent power producers) and 85% (Electrical Equipment & Machinery)[35]. Thus, different scope 3 categories appear to be particularly relevant in certain industries. As of 2017, only a quarter of the companies reporting scope 3 figures within the CDP disclosed emissions for all categories they consider as relevant[35].

In sum, reporting inconsistency, boundary incompleteness, and activity exclusion contribute at different stages to errors in scope 3 emissions measurement. While reporting inconsistency occurs after the accounting process, boundary incompleteness and activity exclusion occur due to misjudgments prior to the actual measurement. As previous literature has discussed the three sources of error independently, our framework aims for completeness. Correcting for the errors in the three areas allows for quantification of omitted scope 3 emissions, as well as for calculating harmonized carbon footprints. Figure 1 illustrates the stepwise approach of the framework. The mathematical

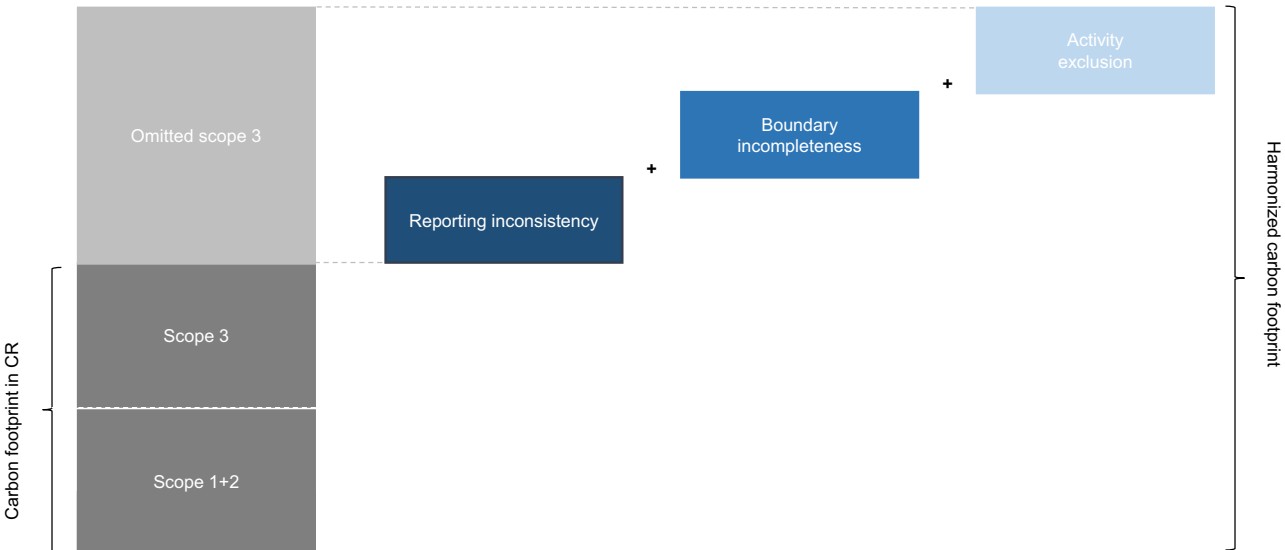

**Fig. 1 Visualization of the framework to harmonize corporate carbon footprints.** The dark gray parts represent the carbon footprint as provided in the corporate report (CR). The blue parts represent potential sources of errors which together form the sum of all omitted scope 3 emissions. The correction of these errors leads to a harmonized carbon footprint.

formulation and a flow chart showing all key input and output flows can be found in the methods section.

To overcome the three sources of error, we analyze each independently to derive the combined effect. Therefore, we resort to information from CRs and the CDP (see supplementary data: sheets 4.1–4.3). CRs include voluntary reports, such as sustainability reports or annual reports, and mandatory reports, such as forms filed for state authorities. They provide information regarding the company's carbon footprint as well as financial and company-related details and may have been prepared in accordance with reporting standards and frameworks, such as the GRI standards, SASB standards, or IR framework. The CDP responses supplement the data basis with more comprehensive environmental information. CDP responses contain emissions figures structured in accordance with the 15 distinct scope 3 categories and provide explanations on the methodology and justifications with regard to missing emission figures.

For reporting inconsistency, we quantify the error by taking the difference between the amount of emissions reported in the CR and in the CDP. We only consider scope 3 emissions since they pose a key challenge—both, in size and complexity. As scope 1 and 2 emissions are mainly calculated using internal data, we assume them in our framework to be reported completely and consistently.

For boundary incompleteness, we classify an emission figure as incomplete in case it does not follow the category-specific minimum boundary of the scope 3 standard in the GHG Protocol (see Table 1). Incomplete boundaries occur, for instance, if only selected means of transportation are included in emissions from business travel or only emissions from first-tier suppliers are included instead of the entire upstream emissions (see supplementary data: sheet 3.1 for case-specific explanations for our case study). To correct incomplete emission figures, we derive category-specific carbon intensities of the peer industry group. Carbon intensities and corrected emission figures are calculated utilizing key performance indicators as emission predictors (see supplementary data: sheet 2.4 and 3.2). We exclude peer companies with incomplete emission figures and use the median to control for outliers. A special case are emission figures subject to incomplete boundaries, but which still show higher intensities than the peer

median. In such cases, we do not adjust the emission figures downwards but keep the self-reported value.

For activity exclusion, an activity is deemed excluded in case the company does not provide an emission figure even though the category is relevant to the business. We assume categories to be relevant unless the company specifically states that emissions are non-existent. All other justification, such as unavailability of data, non-significant amounts of emissions, or the lack of evaluation are not accepted (see supplementary data: sheet 3.1 for case-specific explanations for our case study). This strict approach helps to overcome the challenge posed by the qualitative formulation of the criteria for identifying relevant scope 3 activities in the GHG Protocol. It avoids different interpretations and limits the leeway granted in favor of enhanced comparability. We derive the emissions of excluded scope 3 categories analogous to the calculation of adjusted emissions in case of boundary incompleteness.

**Case study on harmonizing carbon footprints of tech companies.** Tech companies themselves have identified climate change as a key area of concern for their businesses since it poses important social and environmental issues that need to be managed. Several have announced progressive pledges to reduce their greenhouse gas (GHG) emissions and become entirely carbon neutral or even carbon negative[4–7]. In addition to the general ambiguities in carbon disclosures, these climate action ambitions are criticized for a lack of transparency[36].

The amount of energy consumed by tech companies elevates the need for a standardized view on carbon emissions in this sector. With their energy consumption, digital technologies caused 4% of global GHG emissions in 2020, and the share is set to double by 2025[37]. The tech sector consists of industries that are among the highest emitting[35]. With 97% upstream scope 3 emissions, the United States (U.S.) computer manufacturing industry surpasses the industry average of 75%[10,38].

For our case study, we select companies that adhere to the Forbes Global 2000 List 2019. This index ranks the world's largest public companies according to sales, profit, assets, and market value[39]. The focus on public companies offers the advantage of higher data availability. The technology sector in the index is split

into three industries: IT software and service (ITSS), technology hardware and equipment (THE), and semiconductors. To ensure the continued relevance of the sample, we exclude companies which are no longer part of the Forbes Global 2000 List 2020. This results in 55 ITSS companies, 51 THE companies, and 26 semiconductor companies spread across Asia, Europe, and the U.S (see supplementary data: sheet 3.4 for summary statistics). For our case study, we exclude the smallest group, semiconductor companies, since the framework's robustness is linked to the number of comparable peers. The framework set-up requires company-specific information from corporate reports (CRs) and the Carbon Disclosure Project (CDP). Thus, only companies, which submitted a CDP response in 2019 can be considered. Less than half and around two thirds of the companies in the ITSS and the THE sample respectively submitted a valid CDP response in 2019. This results in our final samples with 22 ITSS and 34 THE companies.

For the first source of error, reporting inconsistency, we find lower scope 3 emissions in the CR than in the CDP response for half the tech companies. In the ITSS sample, we find this gap between CR and CDP for 68% of the companies. Thereby, ITSS companies report certain scope 3 categories inconsistently. For instance, five out of the eight companies, that disclose at least some scope 3 emissions in the CR, report emissions from business travel (category 6) and employee commuting (category 7) inconsistently. In the THE sample, 38% of the companies report inconsistently. Nonetheless, it is worth noting that disclosing no scope 3 emissions on either channel results in consistent reporting although full-scale reporting is absent. This applies to five companies in the THE sample but none in the ITSS sample (see supplementary data: 2.3).

For the second source of error, boundary incompleteness, we find that in total, the 56 tech companies report 380 category-specific scope 3 emission figures. Of these 380 figures, we find 15% to be incomplete. Boundary incompleteness applies to 33 companies, 11 from the ITSS and 22 from the THE sample. The extent at the firm level ranges from one to eight incomplete categories and appears particularly often in upstream categories such as *business travel* and *purchased goods and services* (see supplementary data: sheet 2.2 and 3.1 for details).

For the third source of error, activity exclusion, we find 282 excluded categories in total, spread across 18 ITSS and 29 THE companies (see supplementary data: sheet 2.1 and 3.1 for details). The extent of exclusion ranges from neglecting a single category to omitting the entire scope 3. Notably, categories which contribute significantly to total emissions are found lacking (e.g., 30% of the companies neglect purchased goods and services and 43% neglect use of sold products).

In total, we find for our sample of 56 tech companies a gap between reported and harmonized emissions of 391 megatons (Mt) carbon dioxide equivalents ($CO_2e$), of which 202 $MtCO_2e$ originate from omitted upstream and 189 $MtCO_2e$ from omitted downstream emissions. Accounting for these omitted emissions more than doubles self-reported emissions of 360 $MtCO_2e$ to harmonized emissions of 751 $MtCO_2e$. In the following, we present the combined effects on the industry, company, and category level.

On an industry level, emissions levels differ widely between the ITSS and THE industry in absolute terms; companies in the THE sample have eight times higher emissions than in the ITSS sample after the harmonization. Still, the relative gap between self-reported and harmonized emissions appears to be similar. For the ITSS industry, total harmonized carbon emissions nearly double the self-reported figures, which leads to an increase of 39.5 $MtCO_2e$. The increase is based on reporting inconsistency at 60%, boundary incompleteness at 19%, and activity exclusion at 20%.

For the THE industry, total harmonized emissions more than double, with an increase of 351.5 $MtCO_2e$. The increase is based on reporting inconsistency at 31%, boundary incompleteness at 24%, and activity exclusion at 55%. Figure 2 illustrates the results for both samples.

On a company level, the omitted scope 3 emissions are unevenly distributed, both in absolute and relative terms. We find deviations ranging from 0.06% to a factor of 185×, with a quadrupling in the median (see supplementary data: sheet 1.1 for details). This is about twice as high as the increase on industry level, underlines the skewness of the distribution within the sample, and highlights the incomparability of self-reported carbon footprints. In the ITSS sample, almost one third of the companies is subject to omissions in all three areas, another third is subject to two error types. The remainder is affected by one error type. Companies subject to reporting inconsistencies tend to omit a large share of emissions; almost 200% in the median. In cases of boundary incompleteness and activity exclusion, emissions increase in the median by 83 and 117% respectively. For companies from THE sample, 21% are subject to all three error types, and 41% fail on two types (thereof, nearly 60% with boundary incompleteness and activity exclusion). 35% of the companies fall under one type of error (thereof, more than 90% activity exclusion). For THE companies, reporting inconsistency, boundary incompleteness, and activity exclusion increase emissions by 76%, 21%, and 32% respectively in the median. It is noteworthy that additional guidelines do not necessarily prevent scope 3 omissions. ITSS firms that report in accordance with the GRI standards show even higher omissions in the median than firms that do not use or just reference them, while the reverse is true for THE firms. Firms using the IR framework chart a similarly ambiguous picture with fewer omissions in the ITSS sample but more in the THE sample. For both samples, the companies using SASB standards show higher omissions in the median. However, due to their novelty in 2019, SASB standards were only applied by two ITSS and two THE companies and thus the sample might not be representative. Figures 3 and 4 chart the harmonized carbon footprints on company level for both industries and indicate the accordance of the respective CRs with voluntary standards and frameworks.

On a category level, we find that most omitted emissions result from a few dedicated categories. The main part of the increase results from flawed disclosure in the two categories purchased goods and services and use of sold products. Besides these two, only omitted emissions from capital goods contribute a two-digit share with 10% in the ITSS sample. Interestingly, the relative share of the categories remains fairly constant for all three types of error (see supplementary data: sheet 2.1 and 3.1 for comparison). Figure 5 depicts the breakdown by category for both samples.

## Discussion

This paper highlights that current carbon accounting and reporting practices remain unsystematic and not comparable, particularly for emissions along the value chain (scope 3). The framework we present enables the closing of gaps in corporate carbon footprints by accounting for reporting inconsistency, boundary incompleteness, and activity exclusion. We find that companies report different emission levels on different channels, fail to meet the minimum boundaries of emitting activities, or omit relevant scope 3 categories entirely.

In a case study of the tech sector, we find that corporate reports largely understate emissions. By harmonizing scope 3 emissions, we find for a sample of 56 major tech companies a gap between self-reported and harmonized emissions of 391 megatons (Mt)

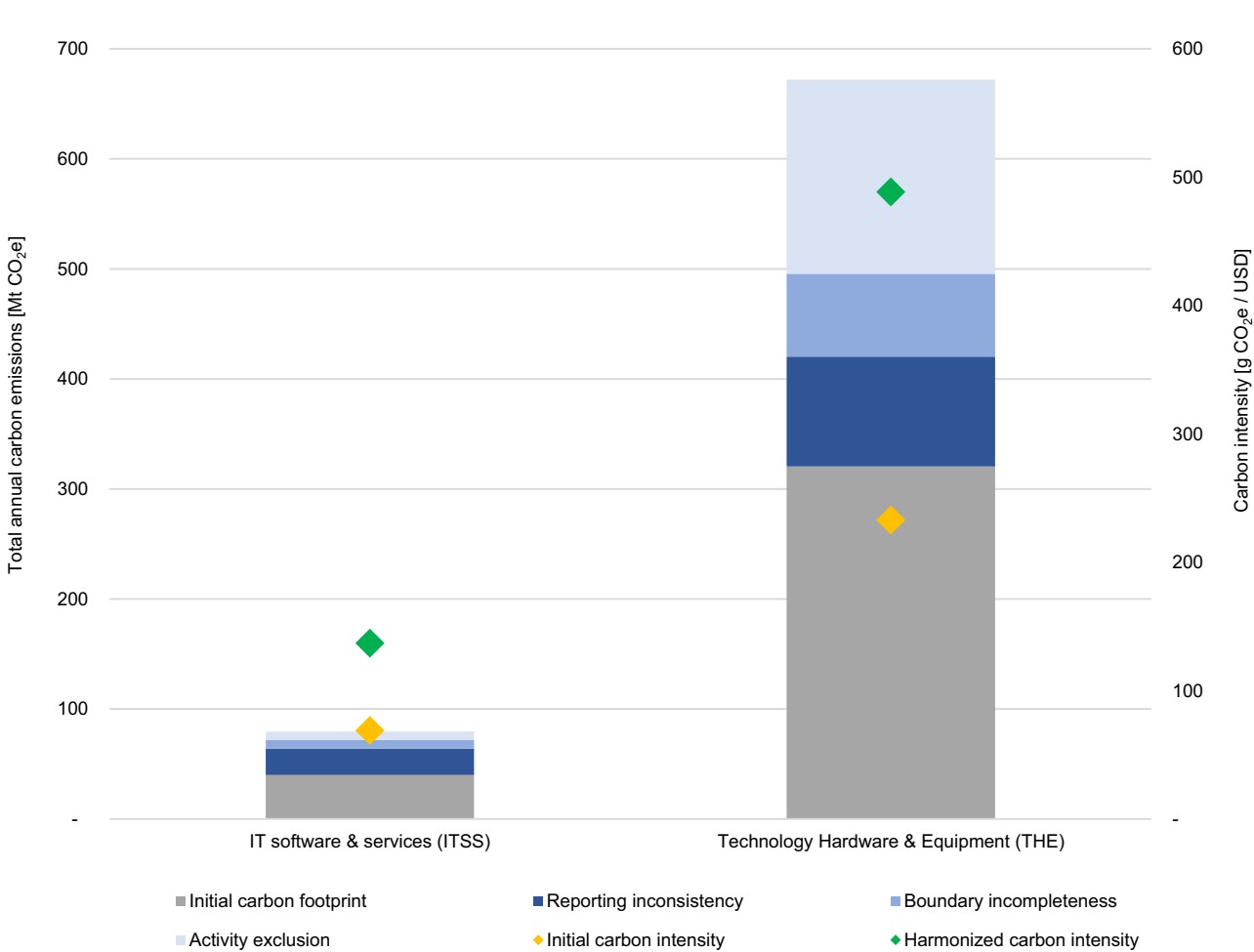

**Fig. 2 Total harmonized carbon emissions of the IT software and service (ITSS) and the technology hardware and equipment (THE) sample in 2019.**
The different sample sizes need to be considered when comparing absolute figures (ITSS: $n = 22$; THE: $n = 34$). The analysis is based on CDP responses of 2019 and corporate reports of the corresponding reporting period. Carbon intensities are calculated by dividing total carbon emissions by total revenues of the sample. See supplementary data: sheet 2.1–2.3 for calculations.

carbon dioxide equivalents ($CO_2$e). Thereof, 202 Mt$CO_2$e originate from omitted upstream emissions and 189 Mt$CO_2$e from omitted downstream emissions, which represents an almost equal contribution to the increase. Interestingly, omitted emissions stem from very few categories which highlights the disproportionate importance of certain scope 3 areas for some industries. Accounting for all omitted emissions more than doubles the amount of self-reported emissions of 360 Mt$CO_2$e to harmonized emissions of 751 Mt$CO_2$e. The size of the gap between self-reported and harmonized corporate carbon footprints suggests a limited consistency in scope 3 emission measurements, which impedes meaningful comparisons. The omitted emissions per annum just from our sample are in the same ballpark as the total annual greenhouse gas (GHG) emissions produced by the nation of Australia[40]. Fortunately, companies with progressive reduction pledges show less discrepancies with a gap of less than 20% (i.e., Microsoft, Google, and Apple).

The case study provides only a snapshot of how reporting inconsistency, boundary incompleteness, and activity exclusion affect corporate carbon footprints. Future research should therefore explore further sectors—and include further companies —to gauge the total gap between self-reported and actual corporate footprints. The oil and gas industry, for instance, poses a particularly interesting case given its high carbon intensity and

recent pledges to move towards net-zero by mid-century[41–43]. A recent Dutch court ruling on Shell underpins the topicality and need for action in this sector[44]. The landmark ruling orders Shell to reduce 45% of emissions by 2030—including scope 3—and holds Shell responsible for up- and downstream emissions[45].

As harmonized carbon footprints are calculated on the basis of peer companies, future research with larger samples as well as longer analysis periods may better control for outliers. Nonetheless, besides the tradeoff between homogeneity and size of the sample, secondary data and adjusted emission figures may never capture all company-specific circumstances. The use of emission predictors and carbon intensities derived from peer companies requires similar expense structures across the sample and underlines the need to analyze industries separately. The challenge of comparability remains as companies may choose different approaches to account for up- and downstream players in different parts of the world. Thus, the calculated emission estimates represent a mix of calculation methods and regional characteristics and cannot fully replace company-specific scope 3 accounting. Still such case studies may provide insights on industry level, and point to gaps in corporate carbon footprints.

Additionally, omitted emissions impede investigating the effectiveness of corporate climate actions on emission reductions. Such transparency, however, is essential to review effectiveness

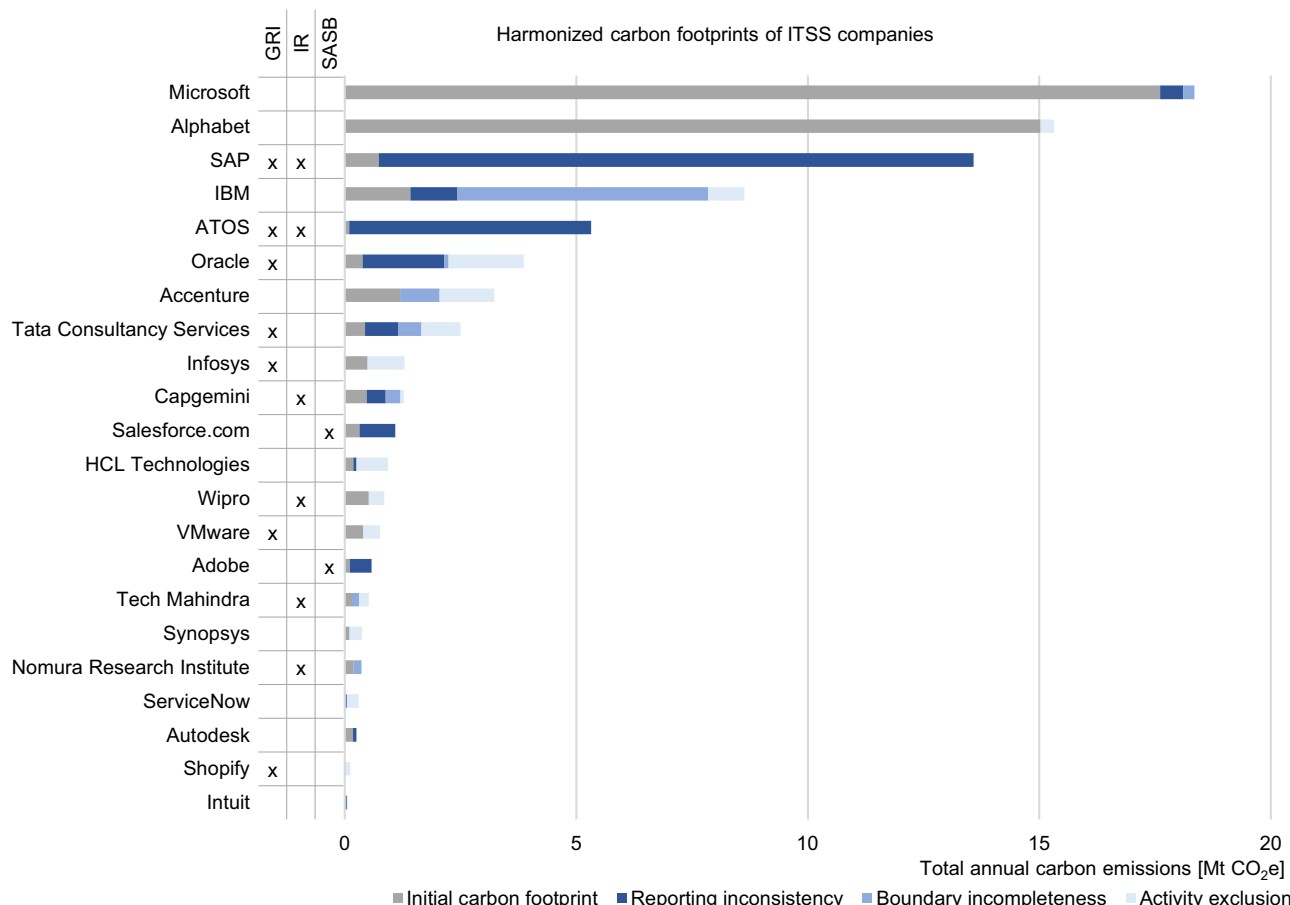

**Fig. 3 Harmonized carbon footprints of IT software and service (ITSS) companies.** Analysis is based on CDP responses of 2019 and corporate reports of the corresponding reporting period. For each company the sum of the initial carbon footprint, as provided in the corporate report, and the omitted emissions form the harmonized carbon footprint. Omitted emissions results from sources of errors such as reporting inconsistency, boundary incompleteness, and activity exclusion. See supplementary data: sheet 2.1–2.3 for calculations. The Global Reporting Initiative (GRI) standards, Integrated Reporting (IR) framework, or Sustainability Accounting Standards Board (SASB) standards are ticked in case the corporate report was prepared in accordance with them.

and improve the design of corporate strategies on the pathway to net-zero emissions. This is important for investors, financial data providers, and policy makers alike. Panel data analyses, for instance, might generate valuable insights to explore the time lag between strategy implementation and visible emission reductions as well as the effect of corporate climate measures. In this context, consistent and complete emission data on company level are required to investigate these relations. Therefore, action to overcome the demonstrated shortcomings appears indispensable.

In light of the current underreporting, it seems unlikely that the current multitude of voluntary guidelines will trigger more accurate carbon disclosure in the future. Standardized and binding regulations with unambiguous guidelines might be more effective. While reporting inconsistency could easily be avoided through obligations to synchronize emission data in corporate reports with any other channel such as the Carbon Disclosure Project (CDP), boundary incompleteness and activity exclusion require more profound advancements.

One option to close the gaps is mandatory regulation for improved full-scale value chain disclosures. In 2019, for instance, the European Union introduced non-binding guidelines for reporting climate-related information, which strongly recommend to disclose scope 3 emissions[46,47]. The guidelines acknowledge the need of comprehensive corporate carbon disclosures and might mark the first step towards binding mandates. Moreover, the

European Commission currently reviews the entire Non-Financial Reporting Directive as part of the action plan on financing sustainable growth, which also includes climate-related information[48]. The public consultations in this context show that more than two-thirds of the users see significant issues with the reliability, comparability, and completeness of the currently reported data, and there is strong support for a requirement on companies to use a common standard[49]. Still, without enhanced digitalization of processes, there is a risk of major inefficiencies in corporate reporting along the supply chain as it requires handling of extensive and complex data. In this context, industry-specific standards which mandate the disclosure of selected scope 3 categories could reduce complexity as well as ambiguity of disclosures.

Binding and internationally standardized scope 1 and 2 emission disclosure may also contribute to close reporting gaps and inconsistencies. Accounting measures today differ among jurisdictions, covering various extents of corporate activities and consequently omitting relevant emissions. The diplomatic and political momentum needed to mandate such standardization, however, has been lacking in the past, and it seems unlikely that all or even a majority of countries will adopt binding reporting guidelines in the near future to correct for the shortcomings, gaps, and ambiguities of existing voluntary guidelines. Even in a scenario with binding reporting guidelines, those would presumably vary greatly across jurisdictions, as seen with other policies and

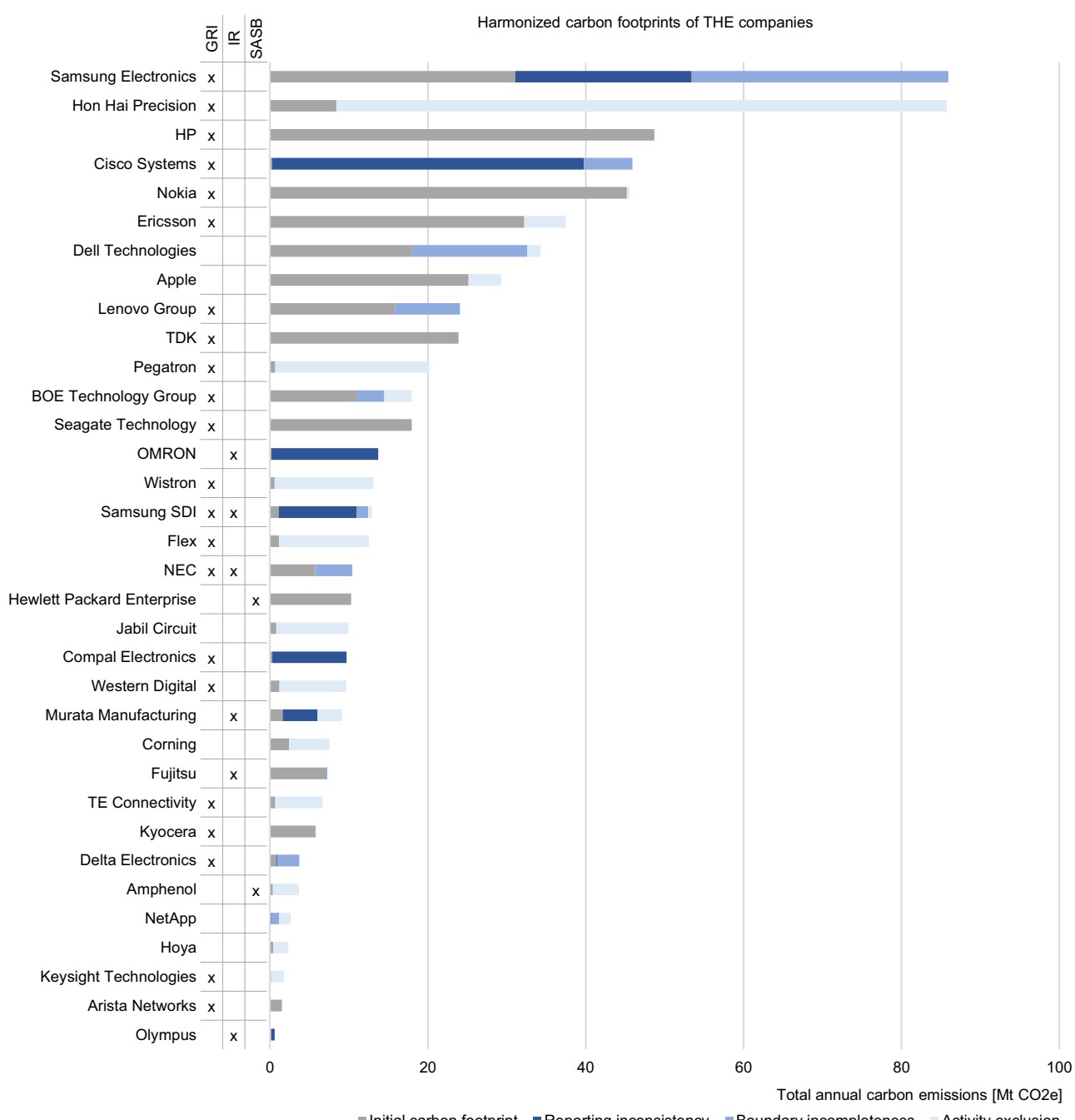

**Fig. 4 Harmonized carbon footprints of technology hardware and equipment (THE) companies.** Analysis is based on CDP responses of 2019 and corporate reports of the corresponding reporting period. For each company the sum of the initial carbon footprint, as provided in the corporate report, and the omitted emissions form the harmonized carbon footprint. Omitted emissions results from sources of errors such as reporting inconsistency, boundary incompleteness, and activity exclusion. See supplementary data: sheet 2.1–2.3 for calculations. The Global Reporting Initiative (GRI) standards, Integrated Reporting (IR) framework, or Sustainability Accounting Standards Board (SASB) standards are ticked in case the corporate report was prepared in accordance with them.

standards. Therefore, improving and consolidating voluntary guidelines appears to be a more realistic option. SASB and IIRC, for instance, merged in June 2021 to form the Value Reporting Foundation[50], and CDP, GRI, SASB, IIRC, and others have announced to seek closer collaboration to improve current guidelines[51]. Also, hybrid approaches aligning voluntary guidelines and global standardization through the International Organization for Standardization (ISO) or the International

Financial Reporting Standards (IFRS) could facilitate the pathway to harmonized domestic standards as well as international policy implementation.

Besides transparency for external stakeholders, binding mandates for scope 1 and 2 can also yield emission reductions without a negative effect on financial performance, as initial empirical evidence from the United Kingdom indicates[52,53]. Additionally, this would make it easier for companies to add up scope 1 and 2

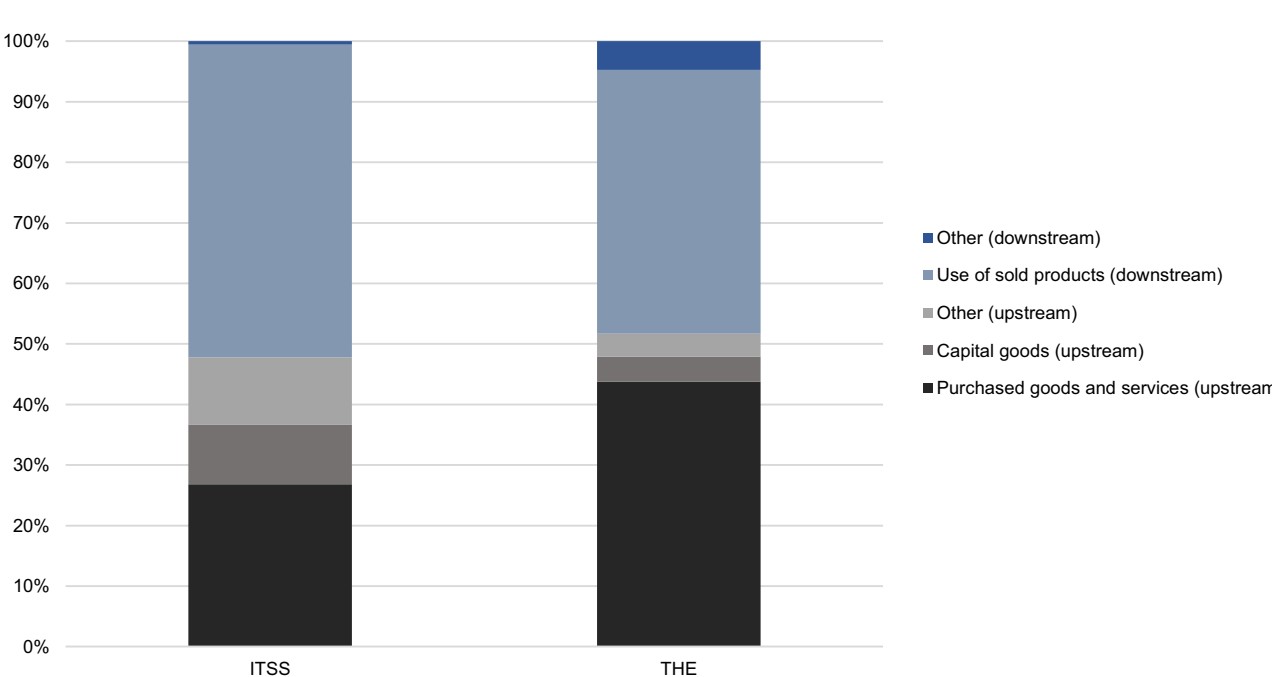

**Fig. 5 Distribution of omitted emissions by scope 3 category of the IT software and service (ITSS) and the technology hardware and equipment (THE) sample.** Analysis is based on CDP responses of 2019 and corporate reports of the corresponding reporting period. See supplementary data: sheet 2.1-2.3 for calculations.

emissions of all suppliers in order to obtain their scope 3 emissions. Binding scope 1 and 2 emission disclosure would furthermore facilitate effective border carbon adjustments[54]. Scope 3 emissions may partly be interpreted as the outsourced environmental damage, and even within the same industry, relative scope 1 and 2 emissions can vary significantly if carbon-intensive activities are shifted to external suppliers[55]. A topical example is the outsourcing of IT infrastructure to cloud service providers[56]. Preventing carbon leakage to jurisdictions with less stringent climate policy regimes calls for transparency on corporate carbon footprints and product embedded emissions.

## Methods

This section provides the formulas to harmonize a company's carbon footprint by quantifying omitted scope 3 emissions. The total carbon footprint is calculated from the sum of the three emission scopes.

$$CF_{Harmonized} = E_{Scope\ 1} + E_{Scope\ 2} + E_{Scope\ 3_{Total}} \quad (1)$$

with:

$CF_{Harmonized}$ = harmonized carbon footprint [$tCO_2e$]
$E_{Scope\ 1}$ = scope 1 emissions [$tCO_2e$]
$E_{Scope\ 2}$ = scope 2 emissions [$t\ CO_2e$]
$E_{Scope\ 3_{Total}}$ = total scope 3 emissions [$t\ CO_2e$]

This framework focuses on scope 3 emissions and thus assumes scope 1 and 2 emissions to be complete and consistently reported across communication channels. Total scope 3 emissions are composed of the emissions reported in the corporate report (CR) and the omitted emissions.

$$E_{Scope3_{Total}} = E_{Scope3_{CR}} + E_{Scope3_{Omitted}} \quad (2)$$

with:

$E_{Scope3_{Total}}$ = total scope 3 emissions [$t\ CO_2e$]
$E_{Scope3_{CR}}$ = scope 3 emissions reported in CRs [$tCO_2e$]
$E_{Scope3_{Omitted}}$ = omitted scope 3 emissions [$t\ CO_2e$]

Figure 6 gives an overview of the framework to calculate the omitted emissions with key input and output flows.

Omitted scope 3 emissions are defined as the sum of reporting inconsistency (RI), boundary incompleteness (BI), and activity exclusion (AE).

$$E_{Scope\ 3_{Omitted}} = E_{Scope\ 3_{RI}} + E_{Scope\ 3_{BI}} + E_{Scope3_{AE}} \quad (3)$$

with:

$E_{Scope\ 3_{RI}}$ = omission due to reporting inconsistency [$t\ CO_2e$]
$E_{Scope\ 3_{BI}}$ = omission due to boundary incompleteness [$t\ CO_2e$]
$E_{Scope3_{AE}}$ = omission due to activity exclusion [$t\ CO_2e$]

**Reporting inconsistency**. Reporting inconsistency is observable in a scenario in which a company is reporting different levels of scope 3 emissions across communication channels. We calculate the difference by deducting the amount of scope 3 emissions reported in the CR from the amount of scope 3 emissions reported in the Carbon Disclosure Project (CDP). The framework does not allow for negative values for reporting inconsistency. For cases in which scope 3 emissions in the CR are higher than in the CDP response we set reporting inconsistency to zero since we assume CDP data to be generally more comprehensive.

$$E_{Scope3_{RI}} = E_{Scope3_{CDP}} - E_{Scope3_{CR}}, s.t. E_{Scope3_{RI}} \geq 0 \quad (4)$$

with:

$E_{Scope\ 3_{CDP}}$ = scope 3 emissions reported in CDP [$t\ CO_2e$]

$E_{Scope\ 3_{CR}}$ = scope 3 emissions reported in CR [$t\ CO_2e$]

**Boundary incompleteness**. We define a scope 3 category as incomplete if the respective minimum boundary described in the GHG Protocol (see Table 1) is not met. We adopt the classification of the 15 distinct scope 3 categories used by the CDP and originally proposed by the GHG Protocol[19]. The sum of all complete scope 3 categories constitutes the total scope 3 emissions.

$$E_{Scope3_{Total}} = \sum_{i=1}^{15} e_i \quad (5)$$

with:

$e_i$ = emissions of scope 3 category i [$t\ CO_2e$]
i = scope 3 category type (1 = purchased goods and services, 2 = capital goods, … , 15 = investments)

To recalculate adjusted values for incomplete emission figures, we derive category-specific carbon intensities of the peer industry group. The carbon

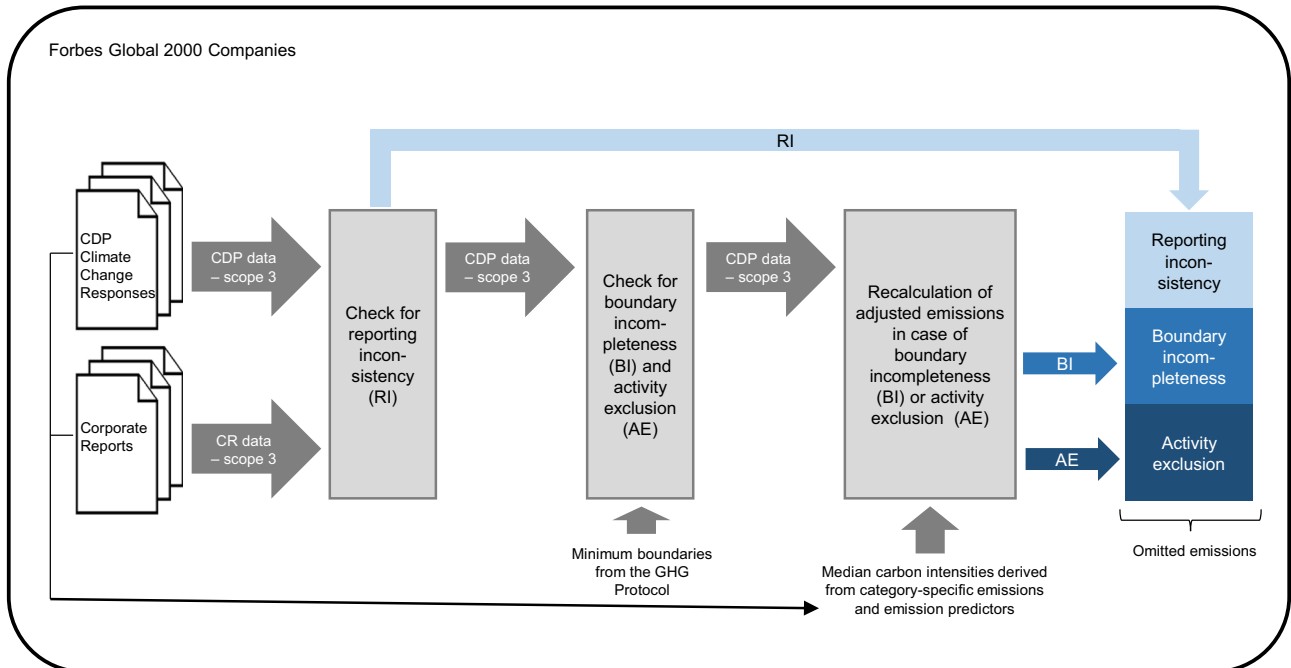

**Fig. 6 Overview of the framework with key input and output flows.** Input data is provided by CDP Climate Change Responses and corporate reports. Throughout the process, the framework checks and adjusts for reporting inconsistency, boundary incompleteness and activity exclusion. The sum of these three sources of errors forms the omitted scope 3 emissions.

intensity of each scope 3 category results from the median of the ratios of the category-specific emissions to the emission predictors across all observed companies. Ratios are only included if the emission figure is above zero and considered complete. Emission predictors vary across scope 3 categories and need to be determined under the constraints of data availability (see supplementary data: sheet 3.2).

$$Order\left(\frac{e_i}{P_i}\right)_j, j = 1, \ldots, N, \, by \, size, \forall \, e_i \, is \, complete \cap e_i > 0 \quad (6)$$

$$I_i = \begin{cases} \left(\frac{e_i}{P_i}\right)_{\frac{N+1}{2}} for \, N \, odd \\ \frac{1}{2}\left[\left(\frac{e_i}{P_i}\right)_{\frac{N}{2}} + \left(\frac{e_i}{P_i}\right)_{\frac{N}{2}+1}\right] for \, N \, even \end{cases} \quad (7)$$

with:

$I_i$ = median carbon intensity of scope 3 category i $[tCO_2e/[P_i]]$.
$P_i$ = emission predictor of scope 3 category i $[[P_i]]$
j = observed peer company (1, ..., N)
    We calculate the adjusted emissions of the incomplete scope 3 categories by applying the respective category-specific carbon intensity to the company's emission predictor.

$$e_{i,adjusted} = P_i * I_i \quad (8)$$

with:

$e_{i,adjusted}$ = adjusted emissions of scope 3 category i $[tCO_2e]$
    The sum of the differences between the adjusted emissions and the initially reported emissions over all categories represents the omission due to boundary incompleteness.

$$E_{Scope3_{BI}} = \sum_{i=1}^{15} e_{i,adjusted} - e_{i,initial}, \forall \, incomplete \, e_{i,initial} \quad (9)$$

with:

$e_{i,initial}$ = initial emissions of scope 3 category i $[t \, CO_2e]$

**Activity exclusion**. The exclusion of activities that cause emissions results from the disregard of the entire scope 3 categories. We assume a category to be excluded if the company does not provide an emission figure in the CDP response despite considering the category to be relevant for their business. We derive the added emissions of undisclosed scope 3 categories with the aid of emission predictors

analogous to the calculation of adjusted emissions in case of boundary incompleteness.

$$e_{i,added} = P_i * I_i, \forall \, e_{i,initial} = 0 \, and \, relevant \quad (10)$$

with:

$e_{i,added}$ = added emissions from scope 3 category i $[tCO_2e]$
    The omission due to activity exclusion is the sum of the added emissions of the excluded scope 3 categories.

$$E_{Scope3_{AE}} = \sum_{i=1}^{15} e_{i,added} \quad (11)$$

## Data availability
All data used and generated in this study are available within the Supplementary Data. The data used in this article includes data points from CDP. The reproduction of any part of the CDP data by any third party is forbidden.

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

## Acknowledgements

The authors would like to thank Sabine Englberger, Sebastian Kahlert, and Gunther Glenk for valuable feedback on earlier drafts of the article.

## Author contributions

L.K. conceived the study. L.K. and C.S. contributed to the design of the study. L.K. aggregated and analyzed the data. L.K. and C.S. drafted the manuscript.

## Funding

## Competing interests

The authors declare no competing interests.

## Additional information

**Peer review information** *Nature Communications* thanks Ki-Hoon Lee, Mathilde Fajardy and the other anonymous reviewer(s) for their contribution to the peer review this work. Peer reviewer reports are available.

