## [Peer Review File · Nature Communications]

Reviewer comments, initial round of review: - -

Reviewer #1 (Remarks to the Author):

Corporate carbon emission is the timely and important issue to address for global scientists and business community. This paper addresses challenging issues on corporate carbon emission measures, and contributes to the ongoing debate on corporate carbon footprint by tackling key issues of corporate carbon emissions' scopes, boundaries, and measures. The GHG protocol developed by WBCSD/WRI provides voluntary reporting guidelines on Scope 1, 2 and 3 corporate carbon emissions. The GHG protocol has been widely adopted by many companies to report their carbon emissions, and leading international corporate sustainability/environmental, social and governance (ESG) index (e.g. Bloomberg, MSCI, DJSI) used the protocol to measure corporate carbon emissions.

As the authors pointed out, Scope 3 (value chain of the corporation) of the GHG protocol has been a challenging area to measure and set the boundaries due to the complexity of corporate value chains. Even the Dow Jones Sustainability Index (DJSI)'s champion companies report mainly Scope 1 and 2 for corporate carbon emissions, and acknowledge the difficulties to measure Scope 3 emissions beyond the corporation's direct control (i.e. upstream supply chain, multi-tier suppliers). The proposed framework which the authors provided is to help demonstrate some lacking issues of reporting consistency and boundary completeness. The primary data sets are from corporate report and the Carbon Disclosure Project (CDP). The CDP provides some useful and comprehensive information on each corporate carbon emission activities, most information are voluntary and qualitative. As the authors argued, voluntary reporting has some limitations to measure and report for full scales of carbon emissions. Some important issues not included in this paper is about measurement approaches. Under the GHG Protocol, accounting measures are adopted to measure corporate carbon emissions. Existing accounting measures may miss some important corporate activities-related carbon emissions, which possibly lead to omissions of carbon emissions in the entire corporate carbon footprint. Even in corporate value chain, this issue can be much complex due to different measures and accounting tools in different countries. Without standardizing corporate carbon emission measures, the issues of reporting consistency and boundary incompleteness seem to continue. New tools such as carbon accounting or environmental management accounting can help to improve corporate carbon measures and reporting activities.

For corporate carbon reporting, I would like to bring two voluntary reporting guidelines (Global Reporting Initiatives and Integrated Reporting) to complement the GHG protocol. Global Reporting Initiatives (GRI) is the first corporate sustainability reporting guideline globally early 1990s to promote sustainability reporting. More recently International Integrated Reporting (IR) Council provides IR framework and guideline for corporate financial and non-financial disclosure. Both corporate reporting guidelines cover corporate carbon footprint measures for consistency, reliability and completeness. For example, GRI provides reporting guidelines for supply chain-focused carbon measures and IR provides capital-based corporate carbon measures. Relating to Consistency and Reliability, IR requires the information on a basis that is consistent over time; and in a way that enables comparison with other organizations to the extent it is material to the organization's own ability to create value over time.

The GHG protocol has been useful to guide many companies to measure and report their carbon emissions, two other reporting guidelines (GRI and IR) also complement the GHG protocol to improve reporting consistency, reliability and completeness.

The case study of tech companies also provides some snapshot of reporting inconsistency, boundary incompleteness and activity exclusion. As the case study used relatively short periods between July 2017-March 2019, some careful interpretations are required. Because any corporations have strategies to operate in multi-markets and regions, the required resources and commitments to implement business strategies directly and indirectly relating to carbon emissions can vary. Also business environment may affect business operations to reduce or increase

corporate carbon emissions, it would be worthy to see longer time period or panel data analysis to see clear relations between corporate activities and corporate carbon emissions with key variables, possibly to affect the relationships between the two.

Overall, this paper contributes to the debate on corporate carbon emissions in a timely manner. Importantly this paper points out lacking areas of the GHG protocol and related measurement scopes and items. It would be beneficial to wider audience and communities if the authors can compare the GHG protocol with complementary guidelines (e.g. GRI guideline, IR guideline) and international sustainability index (e.g. Bloomberg, MSCI, DJSI), and provide strong recommendations for corporations as well as regulatory agency/government to improve carbon emission measure and reporting for transparency.

Reviewer: Professor Ki-Hoon Lee

Reviewer #2 (Remarks to the Author):

With their manuscript, the authors provide a timely and very well-written assessment of important shortcomings in the mainstream methodology used to estimate supply chain emissions. They do so by developing a harmonized framework to quantify supply chain emissions and identify inaccuracies in corporate supply chain emissions reporting by accounting that are owed to three factors: reporting inconsistency, boundary incompleteness, and activity exclusion. They then proceed to demonstrate their analytical framework by applying it to a set of 56 companies in the IT software and service as well as the technology hardware and equipment sectors. The striking finding of this analysis is that corporate reports, on average, omit half of total emissions.

Not only does this manuscript add to a still underserved area of academic research, but its practical relevance is substantial in light of the recent and dramatic surge of corporate commitments to achieve net zero emissions. Substantial confusion about conceptual and methodological aspects of emissions measurement and target-setting are already proving to be meaningful obstacles to the credibility and robustness of such corporate initiatives, casting doubt and uncertainty on their real climate contribution in the eyes of policy makers and other stakeholders (such as investors and the broader public), and threatening to either obfuscate the need for mandatory government policies or undermine public acceptance of and support for voluntary corporate sustainability efforts.

Methodologically, the manuscript adopts an ingenious approach, building on previous peer-reviewed research that has identified individual shortcomings of the Scope 3 emissions reporting framework and its application in practice, and bringing these individual approaches together in a holistic framework that makes use of different forms of publicly available data to identify inconsistencies and fill gaps in self-reported data. There are some assumptions that could be quibbled with, e.g. a blanked assumption that Scope 1 and Scope 2 emissions reporting is accurate (p. 6), yet there is also no doubt that, in terms of complexity, Scope 3 emissions reporting raises by far the greatest challenges, and hence is likely to be the largest source of inconsistencies or omissions.

Where some further detail would be helpful is in the description of the Scope 3 emissions reporting categories and guidelines, and further elucidation – potentially with the help of examples – of the three factors contributing to underreporting of emissions. As it is, the explanations remain somewhat sparse, leaving readers unfamiliar with the different reporting standards and rules in a position where they have to rely on the conclusions of the author(s) as opposed to being able to reach the same conclusion based on the presented facts. The more detailed description of each of the three factors in the “Methods” section (pp. 15-18) comes too late in the manuscript to help readers unfamiliar with emissions reporting standards who might be struggling in the earlier sections of the manuscript (pp. 3-6).

Finally, one further comment: while the authors are correct that international harmonization of corporate Scope 1 and 2 emissions reporting obligations would allow to reliably close reporting

gaps and inconsistencies, the diplomatic and political momentum needed to mandate such requirements is lacking. Similarly, it is unrealistic to assume that all or even a majority of jurisdictions will adopt binding reporting guidelines to correct for the shortcomings, lacunae and ambiguities of current voluntary guidelines – and even if they did, the national guidelines would presumably vary greatly across jurisdictions, as is the case with other policies and standards. What would it thus take to overcome the weaknesses of the current voluntary guidelines – without requiring recourse to mandatory government policymaking? Explaining how e.g. WRI and the WBCSD, or the SBTi, or CDP, could improve their existing guidelines might have a greater chance of seeing real-world implementation. Or, alternatively, how about a hybrid between purely private, voluntary rulemaking and global standardization through ISO, which still is private, but often forms the basis for public policies and domestic standards (and tends to ensure greater uniformity across geographies)?

The authors are encouraged to take these thoughts and suggestions under consideration, but the manuscript is nonetheless publishable after relatively minor revisions, including the more detailed comments below.

Detailed comments:

- Throughout: use of “incomparable” as a term to indicate the difficulty of straightforward comparison may confuse some readers; commonly, the word denotes “matchless”, “without equal”, that is, as an adjective denoting a superlative or exceptional status.
- Without further context, footnotes 2 and 3 are not clearly linked to the statement they supposedly back up (e.g. it’s unclear how the SDGs report 2019 increases pressure on corporate actors – the SDGs are directed at states, who in turn should pass policies to operationalize them domestically)
- The tech sector is undoubtedly a pioneer in the recent trend towards carbon neutrality pledges, but even more interesting – given the scale of the challenge – is the recent surge in oil and gas companies pledging net-zero emission targets and including their scope 3 emissions therein, meaning they intend to compensate or offset the emissions associated with the combustion of any remaining fossil fuels sold.
- Pp. 3-4: It remains somewhat unclear why companies have an incentive to underreport emissions in their corporate reports, but not in their reporting to the CDP: assuming readers (including this reviewer) are not familiar with the technical details and guidelines of CDP and CR reporting, it would help to explain better what the CDP scoring is based on if not emissions (since apparently reporting higher emissions does not rule out a favorable score?)
- P. 4: “each of the 15 scope 3 category”  “each of the 15 scope 3 categories”; here, too, familiarity with the Scope 3 reporting methodology and its 15 categories seems to be assumed, but readers unfamiliar with the GHG Protocol Scope 3 supplement would benefit from at least a bit more explanatory detail.
- P. 12, Fig. 5: While this illustration is helpful, the color coding is already difficult to decipher in the on-screen document, and likely to become even more difficult to read in the print version. Bundling some of the less important emission categories might help, or using a greater number of colors to allow easier distinction of neighboring categories (especially those with very low percentages).

Reviewer #3 (Remarks to the Author):

This contribution addresses corporate scope 3 emissions disclosure. In particular, it quantifies potential gaps in scope 3 emissions reporting by isolating reporting inconsistencies (gaps between corporate reports and Carbon Disclosure Project numbers), boundary incompleteness (emissions reporting does not meet the GHG protocol boundary for a given category), and category exclusion (one of the 15 categories of the GHG protocol scope 3 framework is missing yet potentially important). As this study estimates the gaps resulting from boundary incompleteness and category exclusion using category specific mean carbon intensities from peer-industry, it therefore cannot not replace actual company-level scope 3 accounting performed by a third party. However it does allow for the identification of key emissions sources within the value chain and points to the need

for systematic and transparent scope 3 emissions accounting.

I think the angle and findings of the study are novel, and very relevant to the climate-related financial disclosure community and policy makers. Furthermore the paper is well written and the methodology is very clear with a complete and transparent dataset. However, I consider that some revisions in the methodology and discussion are required to improve the clarity and transparency of the manuscript.

My improvement suggestions are not linear but rather tied to some methodology points:

- Emissions predictor: a crucial point of the analysis is that emissions predictors are used to calculate category specific median carbon intensities and estimate missing or incomplete category emissions for a given company. For some predictors this assumes that all companies have the same expense structure. If we take the example of business travels which is predicted by operational expenses (except energy), using total operational expenses as the predictor assumes that the share business travel expenses out of total expenses is the same across companies. If business travels accounts for 5% of operational expenses for company A , but 15% for companies B, C, D and E, then the estimated carbon emissions for company A using median carbon intensity of the others would be overestimated. Could you please justify this assumption?

- Mean carbon intensities: the use of mean carbon intensities obtained from peer-industry is helpful to provide a ballpark value for missing or incomplete data for a given company, but it cannot replace company specific scope 3 emissions accounting. It would be helpful for the reader to emphasize this limitation in the discussion.

- Boundary incompleteness: could you provide more information as to how you use the GHG protocol minimum boundary to decide whether a category is incomplete? In the case of business travels for Microsoft, I understand from the excel that business travels is 'incomplete' as car rentals is not included in accounting? It would be helpful to provide a bit of background and justification on these decisions.

- Category exclusion: same comment on how you decide that an excluded category should have been included in the accounting.

- General methods: A figure summarising the framework with key input and output flows is required in the Methods section so that the accounting framework can be quickly understood by the reader, without having to go through all the Excel tabs (which are nonetheless very helpful and clear).

**Authors' Response to Reviewers on
“Harmonizing Corporate Carbon Footprints”
Ms. Ref. No.: NCOMMS-20-45008-A**

We are indebted to the three reviewers for their valuable feedback. In response to their suggestions, we have thoroughly revised our manuscript: First, we have added a section to provide further background on interconnection between the GHG protocol and complementary reporting standards and frameworks, namely GRI standards, SASB standards and the IR framework. Second, we underpinned our recommendations with further arguments how to improve transparency in the conclusion. Finally, we have included all specific comments raised by the reviewers. We believe that the revisions have resulted in a more focused and accessible Article.

The following points respond to each reviewer's comments in detail. For this purpose, the original reviewer comments have been italicized, while our responses are shown in regular font.

Reviewer #1

Remarks to the Author:

Corporate carbon emission is the timely and important issue to address for global scientists and business community. This paper addresses challenging issues on corporate carbon emission measures, and contributes to the ongoing debate on corporate carbon footprint by tackling key issues of corporate carbon emissions' scopes, boundaries, and measures. The GHG protocol developed by WBCSD/WRI provides voluntary reporting guidelines on Scope 1, 2 and 3 corporate carbon emissions. The GHG protocol has been widely adopted by many companies to report their carbon emissions, and leading international corporate sustainability/environmental, social and governance (ESG) index (e.g. Bloomberg, MSCI, DJSI) used the protocol to measure corporate carbon emissions.

As the authors pointed out, Scope 3 (value chain of the corporation) of the GHG protocol has been a challenging area to measure and set the boundaries due to the complexity of corporate value chains. Even the Dow Jones Sustainability Index (DJSI)'s champion companies report mainly Scope 1 and 2 for corporate carbon emissions, and acknowledge the difficulties to measure Scope 3 emissions beyond the corporation's direct control (i.e. upstream supply chain, multi-tier suppliers). The proposed framework which the authors provided is to help demonstrate some lacking issues of reporting consistency and boundary completeness. The primary data sets are from corporate report and the Carbon Disclosure Project (CDP). The CDP provides some useful and comprehensive information on each corporate carbon emission activities, most information are voluntary and qualitative. As the authors argued, voluntary reporting has some limitations to measure and report for full scales of carbon emissions.

Some important issues not included in this paper is about measurement approaches. Under the GHG Protocol, accounting measures are adopted to measure corporate carbon emissions. Existing accounting measures may miss some important corporate activities-related carbon emissions, which possibly lead to omissions of carbon emissions in the entire corporate carbon footprint. Even in corporate value chain, this issue can be much complex due to different measures and accounting tools in different countries. Without standardizing corporate carbon emission measures, the issues of reporting consistency and boundary incompleteness seem to continue. New tools such as carbon accounting or environmental management accounting can help to improve corporate carbon measures and reporting activities.

Thank you for your comments. We have captured your thoughts together with further reviewer feedback in the section on “Accounting, Reporting, and Comparison of Corporate Emissions” (which also includes the role of sustainability indices), and enriched the conclusions and policy recommendations accordingly (e.g., to capture the need for standardization as you pointed out).

For corporate carbon reporting, I would like to bring two voluntary reporting guidelines (Global Reporting Initiatives and Integrated Reporting) to complement the GHG protocol. Global Reporting Initiatives (GRI) is the first corporate sustainability reporting guideline globally early 1990s to promote sustainability reporting. More recently International Integrated Reporting (IR) Council provides IR framework and guideline for corporate financial and non-financial disclosure. Both corporate reporting guidelines cover corporate carbon footprint measures for consistency, reliability and completeness. For example, GRI provides reporting guidelines for supply chain-focused carbon measures and IR provides capital-based corporate carbon measures. Relating to Consistency and Reliability, IR requires the information on a basis that is consistent over time; and in a way that enables comparison with other organizations to the extent it is material to the organization's own ability to create value over time.

The GHG protocol has been useful to guide many companies to measure and report their carbon emissions, two other reporting guidelines (GRI and IR) also complement the GHG protocol to improve reporting consistency, reliability and completeness.

Thank you for your suggestions. We have highlighted the role of complementary standards and frameworks by including the GRI standards and IR framework accordingly. Furthermore, we have incorporated the SASB standards, which have become increasingly important in the last three years (according to personal communication with carbon accounting and auditing practitioners that we consulted to explore their relevance and handling in practice). For those three standards and frameworks, we have clarified their approaches towards scope 3 emissions in the background section that we have added. Additionally, we indicated in our case study results which companies in our sample apply such standards and frameworks, and how this affects their level of scope 3 omission. We also highlighted the latest developments in the underlying institutions in our conclusion to underpin the discussion of possible future progress (e.g., the recent announcement of CDP, GRI, SASB, IIRC and other key players to intensify collaboration to improve current guidelines).

The case study of tech companies also provides some snapshot of reporting inconsistency, boundary incompleteness and activity exclusion. As the case study used relatively short periods between July 2017-March 2019, some careful interpretations are required. Because any corporations have strategies to operate in multi-markets and regions, the required resources and commitments to implement business strategies directly and indirectly relating to carbon emissions can vary. Also business environment may affect business operations to reduce or increase corporate carbon emissions, it would be worthy to see longer time period or panel data analysis to see clear relations between corporate activities and corporate carbon emissions with key variables, possibly to affect the relationships between the two.

Thank you for your feedback. In response, we added a caveat on the generalizability of the results. We have also captured your point that business strategies and business environments may yield emission reductions with some delay as part of the conclusion section. For such an analysis, consistent and complete emission data on company level are required. Thus, we extended our recommendations on how to close respective emission data gaps (e.g. industry-specific scope 3 mandates, enhanced alignment of standards).

Overall, this paper contributes to the debate on corporate carbon emissions in a timely manner. Importantly this paper points out lacking areas of the GHG protocol and related measurement scopes and items. It would be beneficial to wider audience and communities if the authors can compare the GHG protocol with complementary guidelines (e.g. GRI guideline, IR guideline) and international sustainability index (e.g. Bloomberg, MSCI, DJSI), and provide strong recommendations for corporations as well as regulatory agency/government to improve carbon emission measure and reporting for transparency.

Thank you for your valuable feedback! The new section on "Accounting, Reporting, and Comparison of Corporate Emissions" puts the GHG protocol in context with prominent complementary guidelines and sheds light on the methodology of sustainability indices as provided

by MSCI and S&P. Additionally, we have underpinned our recommendations with further arguments how to improve transparency in the conclusion.

Reviewer #2

With their manuscript, the authors provide a timely and very well-written assessment of important shortcomings in the mainstream methodology used to estimate supply chain emissions. They do so by developing a harmonized framework to quantify supply chain emissions and identify inaccuracies in corporate supply chain emissions reporting by accounting that are owed to three factors: reporting inconsistency, boundary incompleteness, and activity exclusion. They then proceed to demonstrate their analytical framework by applying it to a set of 56 companies in the IT software and service as well as the technology hardware and equipment sectors. The striking finding of this analysis is that corporate reports, on average, omit half of total emissions.

Not only does this manuscript add to a still underserved area of academic research, but its practical relevance is substantial in light of the recent and dramatic surge of corporate commitments to achieve net zero emissions. Substantial confusion about conceptual and methodological aspects of emissions measurement and target-setting are already proving to be meaningful obstacles to the credibility and robustness of such corporate initiatives, casting doubt and uncertainty on their real climate contribution in the eyes of policy makers and other stakeholders (such as investors and the broader public), and threatening to either obfuscate the need for mandatory government policies or undermine public acceptance of and support for voluntary corporate sustainability efforts.

Methodologically, the manuscript adopts an ingenious approach, building on previous peer-reviewed research that has identified individual shortcomings of the Scope 3 emissions reporting framework and its application in practice, and bringing these individual approaches together in a holistic framework that makes use of different forms of publicly available data to identify inconsistencies and fill gaps in self-reported data. There are some assumptions that could be quibbled with, e.g. a blanked assumption that Scope 1 and Scope 2 emissions reporting is accurate (p. 6), yet there is also no doubt that, in terms of complexity, Scope 3 emissions reporting raises by far the greatest challenges, and hence is likely to be the largest source of inconsistencies or omissions.

Thank you for sharing this observation. We fully agree that scope 3 emissions pose the greatest challenges by far. Additionally, a company largely relies on internal data only to measure scope 1+2 (as opposed to scope 3), which significantly raises the chance for a higher quality of reporting. We complemented this point in the manuscript to provide additional background information for this assumption.

Where some further detail would be helpful is in the description of the Scope 3 emissions reporting categories and guidelines, and further elucidation – potentially with the help of examples – of the three factors contributing to underreporting of emissions. As it is, the explanations remain somewhat sparse, leaving readers unfamiliar with the different reporting standards and rules in a position where they have to rely on the conclusions of the author(s) as opposed to being able to reach the same conclusion based on the presented facts. The more detailed description of each of the three factors in the “Methods” section (pp. 15-18) comes too late in the manuscript to help readers unfamiliar with emissions reporting standards who might be struggling in the earlier sections of the manuscript (pp. 3-6).

Thank you for this valuable feedback. Based on your and further reviewer feedback, we added a background section on “Accounting, Reporting, and Comparison of Corporate Emissions”. This includes a table displaying the 15 scope 3 categories and their minimum reporting boundaries. Furthermore, we added more information in the form of examples and elucidations for each of the three contributing factor in the section on “Three Sources for Error and How to Overcome Them”.

Finally, one further comment: while the authors are correct that international harmonization of corporate Scope 1 and 2 emissions reporting obligations would allow to reliably close reporting gaps and inconsistencies, the diplomatic and political momentum needed to mandate such requirements is lacking. Similarly, it is unrealistic to assume that all or even a majority of jurisdictions will adopt binding reporting guidelines to correct for the shortcomings, lacunae and ambiguities of current voluntary guidelines – and even if they did, the national guidelines would presumably vary greatly across jurisdictions, as is the case with other policies and standards. What would it thus take to overcome the weaknesses of the current voluntary guidelines – without requiring recourse to mandatory government policymaking? Explaining how e.g. WRI and the WBCSD, or the SBTi, or CDP, could improve their existing guidelines might have a greater chance of seeing real-world implementation. Or, alternatively, how about a hybrid between purely private, voluntary rulemaking and global standardization through ISO, which still is private, but often forms the basis for public policies and domestic standards (and tends to ensure greater uniformity across geographies)?

Thank you very much for your comment. We captured your thoughts together with our recommendations in the concluding section.

The authors are encouraged to take these thoughts and suggestions under consideration, but the manuscript is nonetheless publishable after relatively minor revisions, including the more detailed comments below.

Detailed comments:

- Throughout: use of “incomparable” as a term to indicate the difficulty of straightforward comparison may confuse some readers; commonly, the word denotes “matchless”, “without equal”, that is, as an adjective denoting a superlative or exceptional status.

Thank you for this hint. We changed the term to “not comparable” to avoid misinterpretations.

- Without further context, footnotes 2 and 3 are not clearly linked to the statement they supposedly back up (e.g. it’s unclear how the SDGs report 2019 increases pressure on corporate actors – the SDGs are directed at states, who in turn should pass policies to operationalize them domestically)

Thank you for sharing this observation. We exchanged the sources to establish a clearer connection to our message.

- The tech sector is undoubtedly a pioneer in the recent trend towards carbon neutrality pledges, but even more interesting – given the scale of the challenge – is the recent surge in oil and gas companies pledging net-zero emission targets and including their scope 3 emissions therein, meaning they intend to compensate or offset the emissions associated with the combustion of any remaining fossil fuels sold.

Thank you for your thoughts. We share your opinion that oil and gas companies pose a particular interesting sector for applying our framework. Thus, we included the suggestion as a worthwhile area for further research and underpinned this with the recent landmark ruling of a Dutch court on Shell (that explicitly clarified the responsibility for up- and downstream emissions).

- Pp. 3-4: It remains somewhat unclear why companies have an incentive to underreport emissions in their corporate reports, but not in their reporting to the CDP: assuming readers (including this reviewer) are not familiar with the technical details and guidelines of CDP and CR reporting, it would help to explain better what the CDP scoring is based on if not emissions (since apparently reporting higher emissions does not rule out a favorable score?)

Thank you for your suggestion. In response to your comment, we complemented the paragraph on reporting inconsistency with more information on the target audience of the two mediums to

provide a clearer understanding on the underlying incentive structure to the reader. Moreover, we added more details on the scoring system of the CDP.

- P. 4: “each of the 15 scope 3 category”  “each of the 15 scope 3 categories”; here, too, familiarity with the Scope 3 reporting methodology and its 15 categories seems to be assumed, but readers unfamiliar with the GHG Protocol Scope 3 supplement would benefit from at least a bit more explanatory detail.

Thank you for the valuable comment. We corrected this and added the additional background section as mentioned above.

- P. 12, Fig. 5: While this illustration is helpful, the color coding is already difficult to decipher in the on-screen document, and likely to become even more difficult to read in the print version. Bundling some of the less important emission categories might help, or using a greater number of colors to allow easier distinction of neighboring categories (especially those with very low percentages).

Thank you for these suggestions. We bundled some of the less important categories and updated the figure.

Reviewer #3

This contribution addresses corporate scope 3 emissions disclosure. In particular, it quantifies potential gaps in scope 3 emissions reporting by isolating reporting inconsistencies (gaps between corporate reports and Carbon Disclosure Project numbers), boundary incompleteness (emissions reporting does not meet the GHG protocol boundary for a given category), and category exclusion (one of the 15 categories of the GHG protocol scope 3 framework is missing yet potentially important). As this study estimates the gaps resulting from boundary incompleteness and category exclusion using category specific mean carbon intensities from peer-industry, it therefore cannot not replace actual company-level scope 3 accounting performed by a third party. However it does allow for the identification of key emissions sources within the value chain and points to the need for systematic and transparent scope 3 emissions accounting.

I think the angle and findings of the study are novel, and very relevant to the climate-related financial disclosure community and policy makers. Furthermore the paper is well written and the methodology is very clear with a complete and transparent dataset. However, I consider that some revisions in the methodology and discussion are required to improve the clarity and transparency of the manuscript.

My improvement suggestions are not linear but rather tied to some methodology points:

- *Emissions predictor: a crucial point of the analysis is that emissions predictors are used to calculate category specific median carbon intensities and estimate missing or incomplete category emissions for a given company. For some predictors this assumes that all companies have the same expense structure. If we take the example of business travels which is predicted by operational expenses (except energy), using total operational expenses as the predictor assumes that the share business travel expenses out of total expenses is the same across companies. If business travels accounts for 5% of operational expenses for company A , but 15% for companies B, C, D and E, then the estimated carbon emissions for company A using median carbon intensity of the others would be overestimated. Could you please justify this assumption?*

Thank you for this comment. While your observation is true that the framework assumes similar expense structures within the analyzed industries, we believe that the estimates based on the data of the peer companies of the same industry can provide a valid proxy for incomplete and excluded categories. In response to your comment, we included this limitation in the conclusion and emphasized the need to analyze industries separately.

- *Mean carbon intensities: the use of mean carbon intensities obtained from peer-industry is helpful to provide a ballpark value for missing or incomplete data for a given company, but it cannot*

replace company specific scope 3 emissions accounting. It would be helpful for the reader to emphasize this limitation in the discussion.

Thank you for this feedback. We agree with this point and captured your comment in the conclusion.

- Boundary incompleteness: could you provide more information as to how you use the GHG protocol minimum boundary to decide whether a category is incomplete? In the case of business travels for Microsoft, I understand from the excel that business travels is 'incomplete' as car rentals is not included in accounting? It would be helpful to provide a bit of background and justification on these decisions.

Thank you for your thought-provoking questions. Based on your and further reviewer feedback, we added a background section on "Accounting, Reporting, and Comparison of Corporate Emissions" which includes a table displaying the 15 scope 3 categories and their minimum reporting boundaries. We decide whether boundaries are incomplete or whether activities are excluded by comparing company's CDP responses to the GHG Protocol's minimum reporting boundary (e.g., Microsoft limits its estimation for business travel to air travel only). We assume categories to be relevant unless the company explicitly states that emissions are non-existent. This enables us to overcome the leeway of the rather vaguely formulated criteria for identifying relevant scope 3 categories in the GHG Protocol. To provide additional background on each decision, we added case-specific explanations for each category we consider incomplete or excluded in the supplementary data: sheet 3.1.

- Category exclusion: same comment on how you decide that an excluded category should have been included in the accounting.

Please refer to the previous answer.

- General methods: A figure summarising the framework with key input and output flows is required in the Methods section so that the accounting framework can be quickly understood by the reader, without having to go through all the Excel tabs (which are nonetheless very helpful and clear).

Thank you for this suggestion. We added a respective figure in the methods section to facilitate the clarity of the framework.

In conclusion, we are truly grateful to all three reviewers for their insightful comments and suggestions. We feel that the resulting changes have made the manuscript more accessible and sharpened our message.

Reviewer comments, second round of review: - -

Reviewer #1 (Remarks to the Author):

Dear Authors,

I would like to thank both authors for improving this paper by considering my feedback and suggestions. Three main areas of improvements are made. First, in introduction, clear focus and purpose of GHG Protocol in line with other voluntary reporting standards and protocols (GRI, SASB, IR) are provided with updated literature and information. Corporate carbon footprints are not only corporate economic issue but also environmental and social issues. Clearly this paper is significant in the field of corporate carbon accounting. The focus and scopes are also linked to the section of conclusion and recommendations to reflect several important caveats and future directions.

Second, new section of Accounting, Reporting, and Comparison of corporate emissions add substantial value to demonstrate 15 categories of Scope 3 to give much more details on corporate activities-related carbon emissions. Both authors provided much updated movements and development in voluntary corporate reporting standards and investor-focused indices such MSCI and DJSI. This new section gives clear directions and relevance to challenges of Scope 3 and key context of this paper.

Third, the methodological section captures integrated reporting practices in the sample companies analysis to demonstrate harmonized carbon footprints. The methodological rigor and improved contributions are well addressed.

Overall, this paper contributes significantly to the debate on corporate carbon emissions and measures. Notably this paper introduces new methodological approach as well as insights to address Scope 3 carbon emission omissions highlighting reporting inconsistency, boundary incompleteness and activity exclusion. I consider this paper provides novel ideas and approaches after revision, therefore I am pleased to recommend this paper for publication.

Reviewer #2 (Remarks to the Author):

Thank you for comprehensively addressing all comments in my review, as well as feedback by the other two reviewers. Based on the revised manuscript and detailed responses to reviewers, I can affirm that the manuscript has been substantially improved and is now publishable without further revisions.

Reviewer #3 (Remarks to the Author):

Dear authors,

Thank you for your detailed response. In my opinion, the proposed amendments address all the reviewers' remarks and questions, and I consider this manuscript ready for publication.

All the best

**Authors' Response to Reviewers on
“Harmonizing Corporate Carbon Footprints”
Ms. No.: NCOMMS-20-45008B**

We are indebted to the positive recommendations and final remarks shared by the three reviewers. The following points respond to each reviewer's comments. For this purpose, the original reviewer comments have been italicized, while our responses are shown in regular font.

Reviewer #1:

Dear Authors,

I would like to thank both authors for improving this paper by considering my feedback and suggestions. Three main areas of improvements are made. First, in introduction, clear focus and purpose of GHG Protocol in line with other voluntary reporting standards and protocols (GRI, SASB, IR) are provided with updated literature and information. Corporate carbon footprints are not only corporate economic issue but also environmental and social issues. Clearly this paper is significant in the field of corporate carbon accounting. The focus and scopes are also linked to the section of conclusion and recommendations to reflect several important caveats and future directions.

Second, new section of Accounting, Reporting, and Comparison of corporate emissions add substantial value to demonstrate 15 categories of Scope 3 to give much more details on corporate activities-related carbon emissions. Both authors provided much updated movements and development in voluntary corporate reporting standards and investor-focused indices such MSCI and DJSI. This new section gives clear directions and relevance to challenges of Scope 3 and key context of this paper.

Third, the methodological section captures integrated reporting practices in the sample companies analysis to demonstrate harmonized carbon footprints. The methodological rigor and improved contributions are well addressed.

Overall, this paper contributes significantly to the debate on corporate carbon emissions and measures. Notably this paper introduces new methodological approach as well as insights to address Scope 3 carbon emission omissions highlighting reporting inconsistency, boundary incompleteness and activity exclusion. I consider this paper provides novel ideas and approaches after revision, therefore I am pleased to recommend this paper for publication.

Thank you very much for your positive recommendation. Your extensive feedback and suggestions helped us immensely in getting the paper to this point.

Reviewer #2:

Thank you for comprehensively addressing all comments in my review, as well as feedback by the other two reviewers. Based on the revised manuscript and detailed responses to reviewers, I can affirm that the manuscript has been substantially improved and is now publishable without further revisions.

Thank you very much for your positive recommendation. Thank you again for the thought-provoking comments and valuable feedback during the review process.

Reviewer #3:

Dear authors,

Thank you for your detailed response. In my opinion, the proposed amendments address all the reviewers' remarks and questions, and I consider this manuscript ready for publication.

All the best

Thank you very much for your positive recommendation. We highly appreciate your excellent comments during the review process.